# Reduction in RF Loss Based on AlGaN Back-Barrier Structure Changes

**DOI:** 10.3390/mi13060830

**Published:** 2022-05-26

**Authors:** Yi Fang, Ling Chen, Yuqi Liu, Hong Wang

**Affiliations:** 1The Engineering Research Center for Optoelectronics of Guangdong Province, School of Physics and Optoelectronics, South China University of Technology, Guangzhou 510640, China; 201920128602@mail.scut.edu.cn (Y.F.); 201920128597@mail.scut.edu.cn (L.C.); phyqliu@mail.scut.edu.cn (Y.L.); 2The Guangdong Provincial Engineering Laboratory for Wide Bandgap Semiconductor Materials and Devices, School of Electronics and Information Engineering, South China University of Technology, Guangzhou 510640, China; 3Zhongshan Institute of Modern Industrial Technology, South China University of Technology, Zhongshan 528437, China

**Keywords:** HEMT, AlGaN, back-barrier, RF loss

## Abstract

We designed a high electron mobility transistor (HEMT) epitaxial structure based on an AlGaN/GaN heterojunction, utilizing Silvaco TCAD, and selected AlGaN with an aluminum composition of 0.1 as the back-barrier of the AlGaN/GaN heterojunction. We enhanced the confinement of the two-dimensional electron gas (2DEG) by optimizing the structural parameters of the back barrier, so that the leakage current of the buffer layer is reduced. Through these optimization methods, a lower drain leakage current and a good radio frequency performance were obtained. The device has a cut-off frequency of 48.9 GHz, a maximum oscillation frequency of 73.20 GHz, and a radio frequency loss of 0.239 dB/mm (at 6 GHz). This work provides a basis for the preparation of radio frequency devices with excellent frequency characteristics and low RF loss.

## 1. Introduction

Devices based on gallium nitride (GaN) have great potential in high-power and high-frequency applications, due to the superior material properties of GaN such as a wide bandgap, high electron mobility, high saturation velocity, high breakdown electrical field, high thermal conductivity and radiation resistance properties [1,2,3,4]. High electron mobility transistors (HEMTs) in GaN devices is a hotspot for research nowadays. The AlGaN/GaN heterojunction is the mainstream structure of a GaN HEMT device. Two-dimensional electron gas (2DEG), which has significantly higher mobility and saturation velocity than bulk electrons, is generated at the AlGaN/GaN heterojunction interface with high areal density (1 × 10^13^ cm^−2^) and without any external doping [4,5]. Therefore, the AlGaN/GaN HEMT device exhibits great saturation current density and radio frequency (RF) power output capabilities. 

The performance of the GaN HEMT device is largely determined by its substrate, and the two most commonly used substrates for it are silicon carbide (SiC) and silicon (Si). The crystal quality of GaN thin film grown on SiC substrate is quite excellent. However, the cost of the SiC substrate is high, and the wafer size is small (3 to 4 inches), which are obstacles to the industrialization of the GaN HEMT device [6,7]. Compared with SiC substrate, Si substrate has a low price and a large wafer size (≥6 inches), which means that a GaN HEMT device based on Si substrate has some advantage in industrialization. However, the Si-based GaN HEMT device also meets some challenges. First, there is a large lattice mismatch and a thermal mismatch between Si and GaN, which will cause cracks and large warpages on the wafer surface [8]. Second, the compatibility issue between the CMOS process and GaN devices’ preparation process has not been resolved [9,10]. For the GaN HEMT RF device, there is still a very serious challenge, which is that the conductive layer is more likely formed at the interface between the Si substrate and the nucleation layer. The GaN HEMT device usually introduces an AlN nucleation layer between the Si substrate and the GaN layer, but its Al atoms will diffuse on the surface and on the inside of the Si substrates to form a conductive layer, which will cause RF loss in the devices working at high frequency and results in limiting output power and efficiency [11]. According to reports, wafer warpage caused by mismatch is greatly improved by adding a suitable Al(Ga)N stress control layer [12], and gold-free ohmic contacts technology has achieved a breakthrough in the compatibility issue mentioned above [13]. However, the problem of RF loss has become an urgent problem to be solved at present. Optimizing the parameters of the substrate and epitaxial structure to suppress the RF loss is key to improving the performance of the Si-based GaN device. In the past ten years, researchers have made a lot of progress in reducing RF loss, such as by etching trenches between the coplanar waveguide (CPW) conductors (Marti et al. 2010) [14], using a low-temperature aluminum nitride (AlN) nucleation layer (Cordier et al. 2017) [15], and Si substrate nitridation (Wei et al. 2020) [11]. They all reduce the RF loss by changing the process conditions, while there are not many reports on the reduction in the RF loss based on the optimization of the buffer layer structure parameters, and this is the focus of our work. We studied the influence of some methods that have been proven to improve the high resistance (semi-insulating property) of the GaN HEMT device by reducing the RF loss [16,17,18].

We designed the GaN HEMT epitaxial structure using simulation software (from Silvaco TCAD). The performance of the device was improved by adopting the back barrier, optimizing the thickness of the buffer layer, and introducing P-type doping. The S parameters of the CPW made on our buffer layer structure show that this work reduces the RF loss of the buffer layer. Compared with traditional GaN HEMT devices, the S_21_ (at 6 GHz) of our device decreased from 1.318 dB/mm to 0.239 dB/mm, a decrease of 81.8%, and the drain leakage current was reduced to 7.0 × 10^−10^ mA/mm.

## 2. Structure Description

Our structure of AlGaN/GaN-based HEMT is shown in Figure 1a. From bottom to top are a high-resistance Si substrate, a buffer layer 1000 nm thick, a GaN channel layer 150 nm thick, and an AlGaN barrier layer 25 nm thick. We used a high-resistance silicon substrate with a resistivity of 5000 Ω·cm in the simulation. We set the Al component to 0.25, which is the most suitable choice derived from our experiments, in the AlGaN barrier layer. We chose the widely used straight gate as the electrode structure of the device we studied. We set the gate length (Lg) to 400 nm and the lengths of the source and drain both to 500 nm. The length of the gate to the drain terminal (Lgd) was 2600 nm, the gate to the source (Lgs) was 1500 nm, and the source to drain (Lds) was 4500 nm.

We first set the drain-source voltage (V_D_) to 1 V and set the gate-source voltage (V_G_) range from −10 V to 1 V to obtain the drain current (I_D_)-V_G_ characteristics (transfer characteristics). Then, we plotted the drain currents against the drain-source voltage V_G_ (output characteristics) for gate bias V_G_ of 0 V. In small-signal feature extraction, drain-source voltage V_D_ can be divided into DC component (V_D-DC_) and AC component (V_D-AC_), and V_D-AC_ is a small signal relative to V_D-DC_. We set V_G_ = −2 V, V_D-DC_ = 4 V, and set the frequency of V_D-AC_ to sweep from 1 GHz to 300 GHz to obtain the frequency characteristics of the device.

To test the RF loss of the epitaxial materials, the conductive part of the HEMT device needs to be removed, leaving only the buffer layer, and the CPW electrodes are fabricated directly on the buffer layer, as shown in Figure 1b. The S_21_ of the CPW is usually used to characterize the RF loss of the buffer layer [19,20]. We used Silvaco to establish the CPW transmission line model. We set the width of the signal electrode to W = 25 μm, the width of the ground electrodes to Wg = 40 μm, the distance between the electrodes to S = 15 μm, and set the thickness of all electrodes to 0.5 μm.

The simulation of this article mainly uses the ATLAS device simulation framework in commercial software Silvaco TCAD. It can obtain the electrical characteristics of the terminal, as well as the internal concentration distribution, the potential distribution, the current density, etc. In order to avoid the transmission loss of microwave signal energy caused by the impedance mismatch of CPW, this design should meet the load impedance matching of the circuit. However, the S-parameters we calculated using ATLAS did not consider impedance matching, but further considered impedance matching using advanced design system (ADS) software that can simulate actual test scenarios for MMIC circuits. We show the computational model constructed by the ADS in Figure 1c,d. We imported the S parameters calculated by the ATLAS (regardless of impedance matching) into the ADS and obtained the S_21_ or loss value of the device and through the simulation of the two-port network.

ATLAS device simulation is based on comprehensive sets of models, including drift-diffusion transport models; energy balance and hydrodynamic transport models; Fermi–Dirac and Boltzman statistics; advanced mobility models; quantum transport models etc. [21,22]. In addition to the above comprehensive sets of models, ATLAS also has many specific built-in models suitable for different scenarios, among which we mainly use the low-field mobility model; the nitride-specific field-dependent mobility model; and the Shockley–Read–Hall (SRH) recombination and the Fermi–Dirac statistics models [23,24]. All of the above models, except SRH, can directly call the default values in the software, and SRH needs to further define each parameter.

SRH is a model for calculating the carrier generation–recombination process. Consider a uniformly doped semiconductor with carrier concentrations *n* and *p* (*n*_0_ and *p*_0_ in equilibrium). Intrinsic carrier concentration and effective intrinsic carrier concentration are denoted by *n_i_* and *n_ie_*, respectively. At equilibrium, the following conditions exist:*n*_0_·*p*_0_ = *n_i_*^2^(1)

When there are traps (or defects) in the semiconductor material, phonon transitions will occur within the forbidden band width. The expression corresponding to the SRH model in ATLAS is as follows:(2)$$R_{SRH}=\frac{pn-n_{ie}^{2}}{TAUP0[n+n_{ie}\exp(\frac{ETRAP}{kT_{L}})]+TAUN0[p+n_{ie}\exp(\frac{-ETRAP}{kT_{L}})]}$$
where *ETRAP* is defined as the difference between the trap level and the intrinsic Fermi level, *T_L_* represents the lattice temperature, and the *TAUN*0 and *TAUP*0 are the electron and hole lifetimes.

We set the electron lifetime as 1 × 10^−7^ s and the hole lifetime as 1 × 10^−7^ s. For GaN materials, the low-field electron mobility mun is set to 900, the low-field hole mobility mup is set to 10, the saturation electron velocity vsatn = 2 × 10^7^, the conduction band density at 300 k nc300 = 1.07 × 10^18^, and the valence band density nv300 = 1.16 × 10^19^; for AlGaN Materials, mun = 600, mup = 10, nc300 = 2.07 × 10^18^, nv300 = 1.16 × 10^19^ [21]; the AlGaN/GaN interface charge density is set to −1 × 10^13^, the electron surface recombination speed is set to 1 × 10^4^, and the hole surface recombination speed is set to 1 × 10^4^.

## 3. Result and Discussion

### 3.1. Buffer Layer Materials

We first investigated the effects of different buffer layer materials on the electrical performance of our HEMT device. We improved the design of the HEMT buffer layer with the selection of a high-resistance Si substrate. First, we chose GaN, AlGaN and AlN as buffer layers to simulate the DC and frequency characteristics. Note that there are four types of AlGaN (the Al components are 0.05, 0.1, 0.15 and 0.2, respectively). The thickness of all materials was set to 1 μm. We set the background carrier concentration of the buffer layer to 1 × 10^14^ cm^−3^. When AlGaN or AlN is selected as the buffer layer material, the back-barrier structure is formed.

Figure 2a shows that the GaN buffer layer produced the highest electron concentration, followed by Al_X_Ga_1-X_N with x = 0.05, 0.1, 0.15, 0.2, and the AlN buffer layer produced the lowest electron concentration. The results of Figure 2a can be explained from two perspectives: on the one hand, the spontaneous polarization is formed between the GaN channel and the Al(Ga)N back barrier with the introduction of the Al component, and the spontaneous polarization direction is opposite to that of the AlGaN/GaN heterojunction, which reduces the polarization effect of the heterojunction, thus reducing the 2DEG induced by it; on the other hand, compared with the GaN channel layer grown on the GaN buffer layer, the thinner GaN channel layer grown on AlGaN is still in a state of compressive strain, and its lattice size is closer to AlGaN’s, thus reducing the degree of lattice mismatch of the subsequently grown AlGaN/GaN heterojunction, which further weakens the piezoelectric polarization effect of the heterojunction [25]. The conduction band distributions of the HEMT devices with different buffer layers are shown in Figure 2b. At the interface between the GaN channel layer and the buffer layer, the introduction of the back barrier significantly improves the conduction band. As the Al component increases in the back barrier, the conduction band increases, deepening the 2DEG potential well at the AlGaN/GaN heterojunction interface and reducing the possibility of thermal electrons spilling into the buffer layer.

As can be seen from Figure 3a, as the Al component in the buffer layer increases, the threshold voltage shows a positive trend on the horizontal axis (the voltage value decreases), indicating that the smaller voltage can pinch off the channel. This corresponds to the positive offset of the peak transconductance on the horizontal axis in Figure 3b. Meanwhile, the peak transconductance decreases with the increase in the Al component. The peak transconductance corresponding to the GaN buffer layer is 231 mS/mm, while that corresponding to AlN is only 173 mS/mm. Figure 3c shows the drain current I_D_ versus V_D_ when the gate bias V_G_ is 0 V. It can be seen that the introduction of the back barrier significantly reduces the maximum saturation current of the device, and when the Al component is higher, the maximum saturation current decreases more obviously. This is because the Al component enhances the parasitic channel effect, thus reducing the electron concentration of the channel layer.

Although the introduction of the back barrier deteriorates the DC characteristics of the HEMT device, its frequency performance is greatly improved. We use ADS to calculate the RF parameters and analyze the RF performance of the HEMT device. Figure 4a shows the frequency performance of the HEMT devices (cut-off frequency *f*_T_ and maximum oscillation frequency *f*_max_) under different buffer layer materials. Figure 4a shows a traditional GaN buffer layer without a back-barrier structure, and 0.05/0.95 represents the device’s buffer layer material as Al_0.05_Ga_0.95_N, and so on. It can be seen that the buffer layer of the back-barrier structure is significantly better than the conventional GaN buffer layer in frequency characteristics. For the AlGaN back barrier, the *f*_T_ and *f*_max_ decrease with an increasing Al component. This may be due to the confinement of the AlGaN back barrier, reduced leakage in the buffer layer, and increased high resistance in the epitaxial layer. The gate length Lg also affects the frequency characteristics of the device. As can be seen in Figure 4a, when the gate length decreased from 500 nm to 200 nm, both *f*_T_ and *f*_max_ of the devices with different buffer layer materials improved. This improvement became more pronounced with increasing the Al component. In general, in order to make HEMT devices suitable for the radio frequency field, Lg must be reduced. However, a more severe short channel effect occurs when Lg is low to a certain value [26,27]. Considering the above situation, we chose a more appropriate 400 nm long straight gate as the gate structure.

As shown in Figure 4b, the RF loss corresponding to the buffer layer with the back-barrier structure is less than that of the traditional GaN buffer layer. For the devices of the AlGaN back-barrier buffer layer, as the Al component decreases, S_21_ decreases and the corresponding RF loss decreases. Due to the optimal RF performance of Al_X_Ga_1-X_N (its Al component is 0.05), we chose it as the buffer layer material in the following simulation.

All the simulation results in Section 3.1 are summarized in Table 1.

### 3.2. Thickness of the Buffer Layer

Next, we investigated the effect of the thickness of the buffer layer on the electrical performance of the HEMT devices. Considering the actual growth of the epitaxial layer, we must control the thickness of the whole epitaxial layer within 2 µm. Therefore, in the following studies, we set the thickness of the buffer layer to be 1 µm, 1.3 µm, and 1.6 µm, respectively.

From Figure 5a, the height of the conduction band increases as the thickness of the buffer layer increases. As can be seen in Figure 5b–d, with the increase in the AlGaN buffer layer thickness, the transfer characteristics of the device do not change significantly, and there are small increases in peak transconductance and output characteristics. The small pictures in Figure 5b,c represent the partial enlargement of the curve (note that the enlarged horizontal and vertical ratio has been adjusted to highlight the characteristics of the curve).

The calculated results of the off-state leakage current of the device are shown in Table 2. The leakage current of the device is approximately 1.8 × 10^−8^ mA/mm at 1 μm and approximately 9.5 × 10^−19^ mA/mm at 1.6 μm, respectively. They illustrate that the increase in thickness significantly reduces the leakage current of the device.

As the thickness of the AlGaN buffer layer increases, the *f*_T_ and *f*_max_ of the device become larger, and the RF performance is improved, as shown in Figure 6. When the thickness was changed from 1 μm to 1.6 μm, the *f*_T_ and *f*_max_ were increased by 9.3% and 4.1%, respectively. In addition, comparing the values of S_21_, we found that the increased thickness within also reduces the RF loss in the scope of our study.

Increasing the thickness is considered another effective way to improve the high resistance of the epitaxial layer. It can be seen from the above results that the high resistance increases as the device thickness increases, which has some effect on reducing the drain leakage and RF loss.

All the simulation results in Section 3.2 are summarized in Table 2.

### 3.3. Doping

Like other GaN materials, there are a lot of impurities in unintentionally doped AlGaN film [28]. In order to form a high-resistance AlGaN buffer layer, its background carrier concentration needs to be reduced. Due to the presence of N-type background carriers in AlGaN, we introduced appropriate P-type polarization doping in the buffer layer to neutralize the N-type background carriers to reduce the background carrier concentration in it. This doping process is simulated by the method of acceptor doping. We set the acceptor trap energy level to 0.36 eV, the trap to capture electrons with a cross-section of 1 × 10^−13^ cm^−2^, and the trapped hole cross-section as 1 × 10^−15^ cm^−2^. We set the doping concentration to 1 × 10^18^ cm^−3^, 1 × 10^19^ cm^−3^, and 1 × 10^20^ cm^−3^, respectively, to simulate the DC and small-signal characteristics of the HEMT device.

Figure 7a shows the electron concentration of the buffer layer at different doping concentrations. As the doping concentration increases, the electron concentration decreases, which may be because, when the device is working in a high electric field condition, the electrons in the 2DEG channel will gain higher energy and overflow to the outside of the channel, and some of them will be trapped by the traps of the AlGaN buffer layer. This corresponds to the reduction in off-state leakage in Table 3. Figure 7b shows the conduction band distribution of the HEMT at different doping concentrations. In the buffer layer, the conduction band increases as the doping concentration of the back-barrier layer increases. It is shown that the introduction of P-type doping effectively strengthens the effect of the back barrier and further suppresses the 2DEG overflow.

As the doping concentration increases, the threshold voltage of the device will drift positive (Figure 8a), and the saturation current will decrease (Figure 8b). The peak transconductance of the device decreases (Figure 8c). These mean that an increase in the doping concentration causes a deterioration of the DC characteristics of the HEMT device, which further confirms the conclusion of Figure 7a.

Figure 9a shows that as the doping concentration increases, the frequency characteristics of the device are improved, and the *f*_T_ and *f*_max_ increase significantly. When the doping concentration reaches 1 × 10^20^ cm^−3^, *f*_T_ reaches 48.9 GHz and *f*_max_ reaches 73.2 GHz. Compared to non-doped devices, the doped devices have an increase of 22.9% for *f*_T_ and 19.4% for *f*_max_. As can be seen from Figure 9b, the S parameter decreases with increasing doping concentration, from 0.641 dB/mm (at 6 GHz) to 0.239 dB/mm (at 6 GHz).

In order to further study the relationship between the doping concentration of the device’s AlGaN buffer layer and the *f*_T_ and *f*_max_ performance of the device, we further studied the variation of the effective capacitance between the device electrodes with the signal frequency. Figure 10 shows the gate-source capacitance (C_gs_) and gate-drain capacitance (C_gd_) with frequency at different doping concentrations of the buffer layer. As the doping concentration increases, C_gs_ and C_gd_ gradually decrease. C_gs_ and C_gd_ can describe the charge–discharge behavior of the depletion layer capacitance with gate voltage. Considering parasitic effects, the formulas describing *f*_T_ and *f*_max_ are as follows:(3)$$f_{T}=\frac{g_{m}/2\pi }{\left(C_{\mathrm{gs}}+C_{\mathrm{gd}})\left[1+(R_{s}+R_{d})/R_{ds}\right]+C_{\mathrm{gd}}g_{m}(R_{s}+R_{d}\right)}$$
(4)$$f_{\max}\approx \frac{f_{T}}{2\sqrt{\frac{R_{g}+R_{i}+R_{s}}{R_{ds}}+2\pi f_{T}R_{g}C_{gd}}}$$
where *g_m_* is the transconductance, *R_g_* and *R_s_* are the gate and source resistances, and *R_i_* and *R_ds_* are the input and output resistances at gate bias. As can be seen from Equations (3) and (4), decreasing *C_gd_* can increase *f_T_*; increasing *f_T_* and decreasing *R_g_*, *R_s_*, *R_i_* and *R_ds_* can increase *f*_max_. This also explains the improvement of *f*_T_ and *f*_max_ with increasing doping concentration of the back barrier.

All the simulation results in Section 3.3 are summarized in Table 3.

To compare the reported losses caused by substrate parasitic conduction, we took advantage of the small-signal CPW line losses reported in the literature, which are summarized in Table 4 below. In contrast, our study employed a back-barrier structure to improve RF loss in the HEMT device, which was not used by others. We obtained the ideal small-signal characteristics and loss values through simulation, which provided the basis for the preparation of the HEMT with low RF loss.

## 4. Conclusions

High-resistance buffer layer material helps to reduce the off-state leakage current of the device and increase the on-off ratio and breakdown voltage. One implementation measure of the high-resistance buffer layer is to introduce an AlGaN back-barrier buffer layer with a wider bandgap. The magnitude of the off-state leakage current is closely related to the RF loss of the material. Therefore, the use of the AlGaN back-barrier buffer layer structure can improve the RF loss of the material to some extent. We found that the most obvious improvement is the introduction of the back barrier, followed by the introduction of P-type doping and increasing thickness.

In conclusion, we employ three methods to enhance the high resistance of the epitaxial layer and study their effects on the DC characteristics, the frequency characteristics, and the RF loss of the device. The simulation results show that the high resistance of the epitaxial layer can significantly improve the frequency characteristics and reduce the RF loss. We have designed a HEMT device suitable for RF, and then on this basis, designed the transmission line model for S-parameter simulation. We used a P-type doped AlGaN back barrier with an Al component of 0.05 and a doping concentration of 1 × 10^20^ cm^−3^ to obtain *f*_T_ of 48.9 GHz, *f*_max_ of 73.20 GHz, and RF loss of 0.239 dB/mm (at 6 GHz). In addition, a simple modeling method for evaluating the RF loss of epitaxial materials is proposed, which facilitates subsequent research.

The results demonstrate that there is a large correlation between the buffer leakage current of the GaN HEMT device and its RF loss and provides new ideas for researchers.

## Figures and Tables

**Figure 1 micromachines-13-00830-f001:**
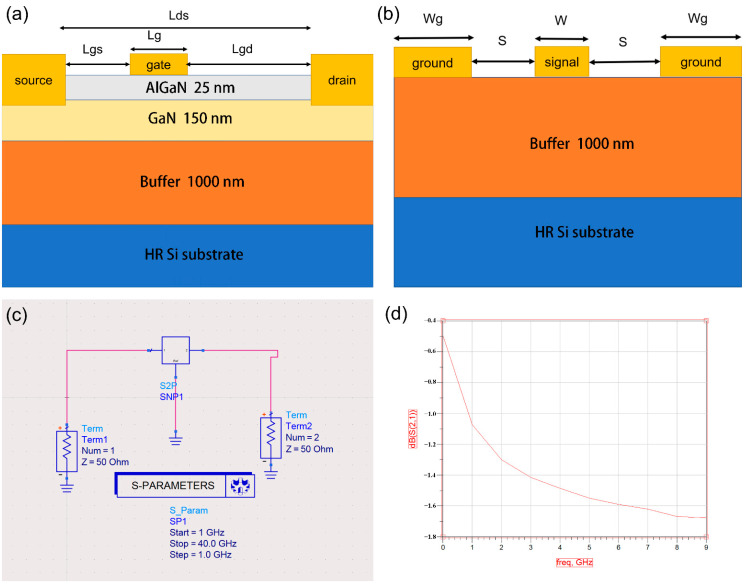
(**a**) The HEMT structure; (**b**) schematic of the CPW structure; (**c**) S-parameter simulation circuit diagram; (**d**) schematic diagram of the S-parameter simulation results.

**Figure 2 micromachines-13-00830-f002:**
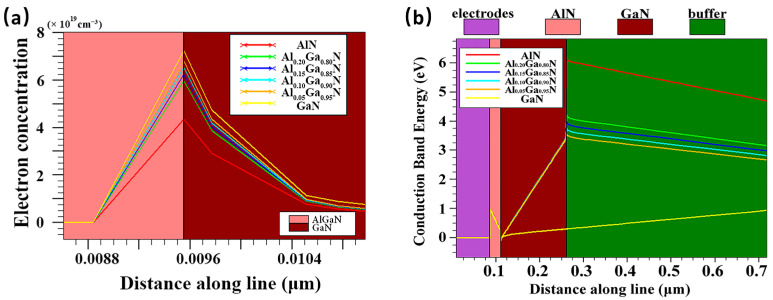
(**a**) Variation of electron concentration at AlGaN/GaN heterojunction interface with different buffer layers; (**b**) Conduction band distribution of HEMT devices with different buffer layers.

**Figure 3 micromachines-13-00830-f003:**
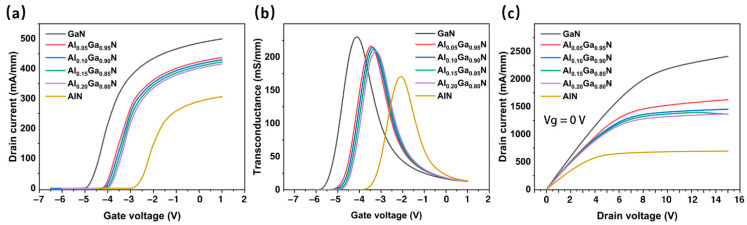
(**a**) Transfer characteristics, (**b**) transconductance, and (**c**) output characteristics of the HEMT with different buffer layers.

**Figure 4 micromachines-13-00830-f004:**
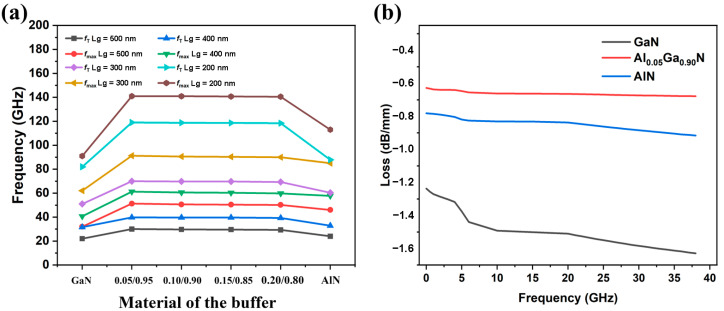
(**a**) The cut-off frequency (*f*_T_) and the maximum oscillation frequency (*f*_max_) of the HEMT buffer layer of different materials and gate lengths. (**b**) The RF losses of the CPW structure with the HEMT buffer layer of different materials.

**Figure 5 micromachines-13-00830-f005:**
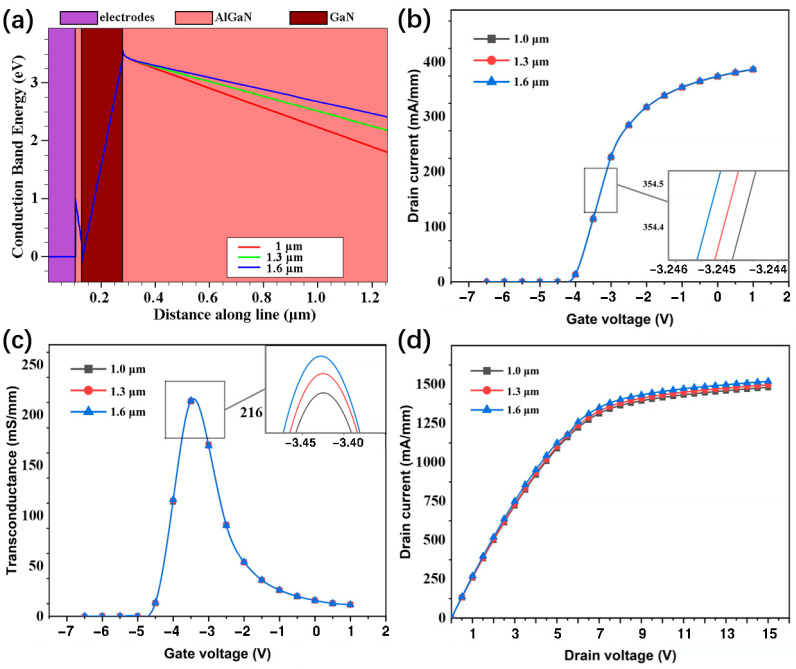
(**a**) Conduction band distributions, (**b**) transfer characteristics, (**c**) transconductance, and (**d**) output characteristics of the HEMT with different thicknesses of AlGaN buffer layer.

**Figure 6 micromachines-13-00830-f006:**
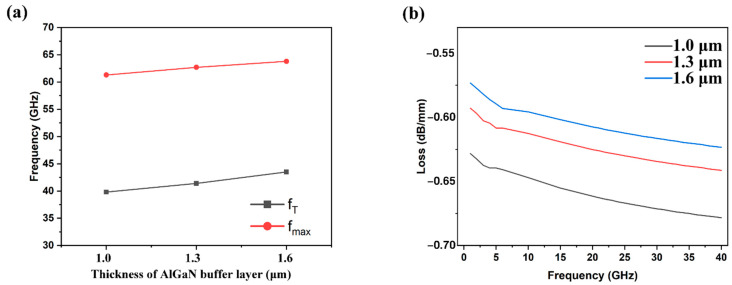
(**a**) *f*_T_ and *f*_max_ of the HEMT with different thicknesses of AlGaN buffer layer; (**b**) the RF losses of the CPW with the HEMT with different thicknesses of AlGaN buffer layer.

**Figure 7 micromachines-13-00830-f007:**
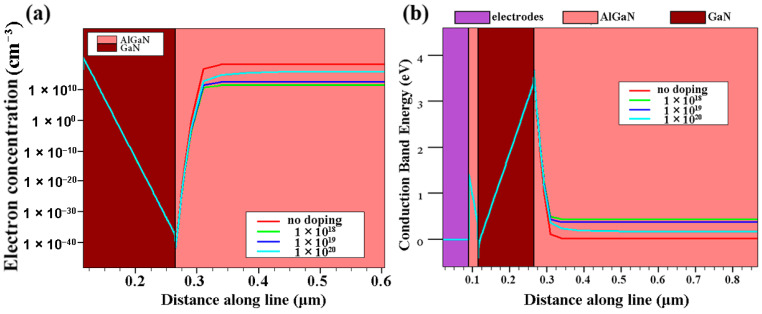
(**a**) Electron concentrations and (**b**) conduction band distributions of HEMT with different doping concentrations of AlGaN buffer layer.

**Figure 8 micromachines-13-00830-f008:**
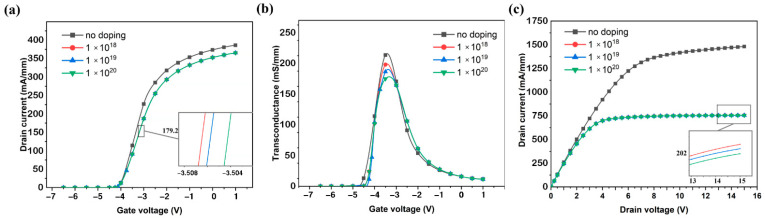
(**a**) Transfer characteristics, (**b**) transconductance, and (**c**) output characteristics of the HEMT of AlGaN buffer layer with different doping concentrations.

**Figure 9 micromachines-13-00830-f009:**
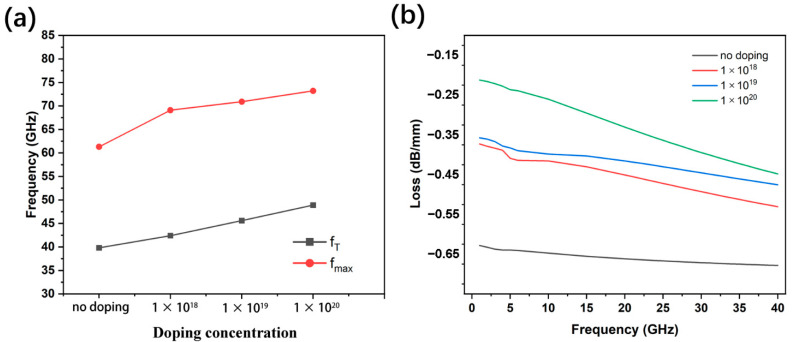
(**a**) *f*_T_ and *f*_max_ of the HEMT of AlGaN buffer layer with different doping concentrations; (**b**) The RF losses of the CPW with the HEMT of AlGaN buffer layer with different doping concentrations.

**Figure 10 micromachines-13-00830-f010:**
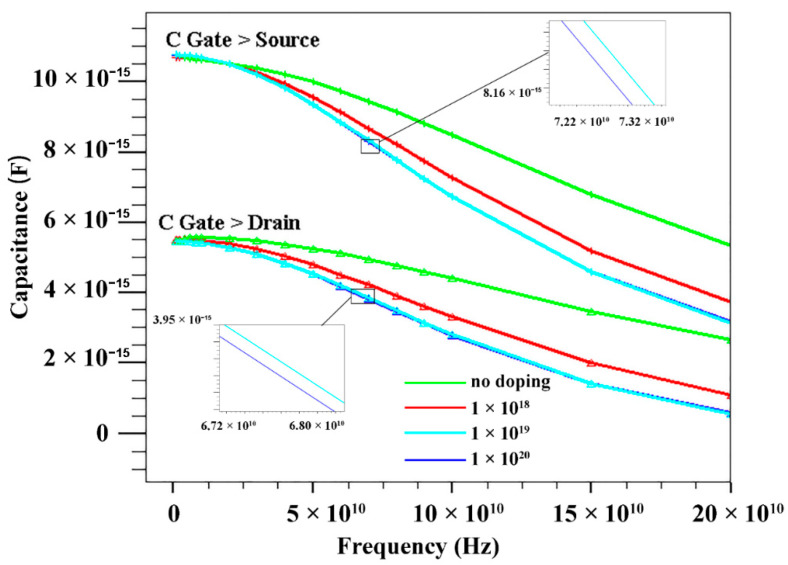
C_gd_ and C_gs_ of HEMT with different doping concentrations of AlGaN buffer layer.

**Table 1 micromachines-13-00830-t001:** DC and AC simulation results of the HEMT with different buffer layers.

Buffer	Threshold Voltage (V)	Peak Transconductance (mS/mm)	Drain Leakage Current (mA/mm)	*f*_T_ (GHz)	*f*_max_ (GHz)
GaN	−4.38	231.3	3.0 × 10^−8^	31.6	40.7
Al_0.05_Ga_0.95_N	−3.62	216.4	2.7 × 10^−8^	39.8	61.3
Al_0.10_Ga_0.90_N	−3.45	212.8	3.5 × 10^−9^	39.7	60.6
Al_0.15_Ga_0.85_N	−3.30	209.1	5 × 10^−9^	39.6	60.4
Al_0.20_Ga_0.80_N	−3.16	205.6	7.8 × 10^−9^	39.3	59.8
AlN	−2.31	173.2	7.1 × 10^−10^	32.8	57.7

**Table 2 micromachines-13-00830-t002:** DC and AC simulation results of the HEMT with different thicknesses of AlGaN buffer layer.

Buffer Thickness (μm)	Threshold Voltage (V)	Peak Transconductance (mS/mm)	Drain Leakage Current (mA/mm)	*f*_T_ (GHz)	*f*_max_ (GHz)
1.0	−3.62	216.4	2.7 × 10^−8^	39.8	61.3
1.3	−3.65	216.6	1.8 × 10^−8^	41.4	62.7
1.6	−3.69	216.8	9.5 × 10^−9^	43.5	63.8

**Table 3 micromachines-13-00830-t003:** DC and AC simulation results of the HEMT of AlGaN buffer layer with different doping concentrations.

Doping Conc (cm^−3^)	Threshold Voltage (V)	Peak Transconductance (mS/mm)	Drain Leakage Current (mA/mm)	*f*_T_ (GHz)	*f*_max_ (GHz)
0	−3.62	216.4	2.7 × 10^−8^	39.8	61.3
1 × 10^18^	−3.60	199.4	3.0 × 10^−9^	42.4	69.1
1 × 10^19^	−3.57	190.3	1.0 × 10^−10^	45.6	70.9
1 × 10^20^	−3.45	178.3	7.0 × 10^−10^	48.9	73.2

**Table 4 micromachines-13-00830-t004:** Epitaxial structure and CPW loss of GaN-on-Si below 20 GHz reported in the literature.

Author	Structure	Loss	Time
Meneghesso G. [29]	Si/(AlGa)N/GaN/AlN	0.9 dB/mm @ 10 GHz	2013
Cao L. [30]	Si/(AlGa)N/GaN/AlGaN	0.58 dB/mm @ 5 GHz	2017
Cordier Y. [15]	Si/(AlGa)N/GaN	0.3 dB/mm @ 10 GHz	2018
Cao L. [20]	Si(HR)/(AlGa)N/GaN/AlGaN	0.27 dB/mm @ 20 GHz	2018
Chandrasekar H. [28]	GaN-on-Si	0.6 dB/mm @ 6 GHz	2019
Wei L. [11]	Si/AlN	1.47 dB/mm @ 6 GHz	2020
Ghosh S. [31]	Si/AlN	12.7 dB/mm @ 5 GHz	2021

## Data Availability

Not Applicable.

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
