# Peer review of "Reduction in RF Loss Based on AlGaN Back-Barrier Structure Changes"

_micromachines, 2022, doi:10.3390/mi13060830_

Round 1

Reviewer 1 Report

The authors have designed a High Electron Mobility Transistor (HEMT) epitaxial structure based on AlGaN/GaN heterojunction utilizing Silvaco TCAD, and select AlGaN with an aluminum composition of 0.1 as the back-barrier of the AlGaN/GaN heterojunction. The simulation results seem promising, but I have a few comments to improve the quality of the manuscript.

  1. The authors mostly investigate the impact of the thickness of the buffer layer, do you foresee any impacts coming from the difference in the gate length? 
  2. Please make a table of comparison highlighting the advantages of this design vs the other published reports in this area.

Reviewer 2 Report

Reference: micromachines-1731935

Review Report

The MS entitled “Reduction of RF Loss Based on AlGaN Back-barrier Structure Changes” by Yi Fang, Ling Chen, Yuqi Liu and Hong Wang reports on a TCAD study of the design of High Electron Mobility Transistors (HEMTs) based on the AlGaN/GaN heterojunction to reduce the leakage current of the buffer layer.

The paper English level is acceptable, but authors must revise the MS carefully.

In my opinion, the paper could be accepted if authors successfully address the questions below:

1 – On lines 121-126 authors give a list of ATLAS capabilities. These lines are irrelevant for the present work and should be removed, but authors must detail the models effectively used in the work. In lines 128-129 the models selected among the available ones are only mentioned, but all the parameters used in each model must also be explicitly given in the paper. The parameter values must be justified (i.e. authors must give the references in literature to the values used).

2 – The vertical band diagram of the structure below the gate must be given.

3 – SRH model was used: could authors detail the parameters’ values used in the model as requested in point 1 above. How was this model implemented across the transistor? I understand that SRH was used to model volume recombination, could authors describe how the recombination at the interfaces was considered?

3 – Why authors didn’t consider including a quantum transport model?

4 – Subsections 3.1 and 3.2 clearly show that the effects of the variation of the buffer layer AlGaN (at least for the range of Al content considered in the TCAD study) and the thickness of the buffer layer are almost negligible (peak gm, fT, …). Please, consider if the extension devoted to these two subsections can be shortened.

5 – On lines 252-254 pointed to a very interesting aspect: “ … some of them will be trapped by the traps of the AlGaN buffer layer, reducing the saturation leakage current and causing threshold voltage drift and other phenomena.” Please, detail the results obtained in the TCAD simulations carried out about charge trapping and discuss them according to points 1 and 3 above.

6 – Could authors give Cgs and Cgd for the 4 levels of doping of the AlGaN buffer layer considered in Table 3?

7 – Finally, could authors detail the procedure followed to pass the data from TCAD to ADS? I guess that ATLAs may provide “h21” that should be enough to obtain fT. What is the interest of using ADS? On Fig. 1 (c) two 50 ohms loas are added to the two ports, but on line 108 it is stated “ … (impedance matching was not considered) into ADS …”. Could authors give more details about that issue and in which conditions fmax was calculated?

Round 2

Reviewer 2 Report

Authors have satisfactorily answered my questions and modified the MS accordingly.

I recommend the MS for publication.